# Oral Nirmatrelvir/Ritonavir Therapy for COVID-19: The Dawn in the Dark?

**DOI:** 10.3390/antibiotics11020220

**Published:** 2022-02-09

**Authors:** Yuan-Pin Hung, Jen-Chieh Lee, Chun-Wei Chiu, Ching-Chi Lee, Pei-Jane Tsai, I-Lin Hsu, Wen-Chien Ko

**Affiliations:** 1Department of Internal Medicine, National Cheng Kung University Hospital, College of Medicine, National Cheng Kung University, Tainan 704, Taiwan; yuebin16@yahoo.com.tw (Y.-P.H.); jclee.eric@msa.hinet.net (J.-C.L.); chichingbm85@yahoo.com.tw (C.-C.L.); 2Department of Internal Medicine, Tainan Hospital, Ministry of Health and Welfare, Tainan 700, Taiwan; bahamudo@gmail.com; 3Department of Medicine, College of Medicine, National Cheng Kung University, Tainan 704, Taiwan; 4Graduate Institute of Medical Sciences, College of Health Sciences, Chang Jung Christian University, Tainan 711, Taiwan; 5Clinical Medicine Research Center, National Cheng Kung University Hospital, College of Medicine, National Cheng Kung University, Tainan 704, Taiwan; 6Department of Medical Laboratory Science and Biotechnology, College of Medicine, National Cheng Kung University, Tainan 704, Taiwan; peijtsai@mail.ncku.edu.tw; 7Department of Pathology, National Cheng Kung University Hospital, Tainan 704, Taiwan; 8Centers of Infectious Disease and Signaling Research, National Cheng Kung University, Tainan 704, Taiwan; 9Department of Surgery, National Cheng Kung University Hospital, College of Medicine, National Cheng Kung University, Tainan 704, Taiwan

**Keywords:** Paxlovid, nirmatrelvir, PF-07321332, ritonavir, SARS-CoV-2, COVID-19, M^pro^

## Abstract

Nirmatrelvir/ritonavir (Paxlovid™) is an effective and safe antiviral drug that inhibits the main protease (M^pro^), 3CL protease, of SARS-CoV-2. A reduction in COVID-19-related hospitalization or death was observed in patients treated with nirmatrelvir/ritonavir within five days of symptom onset. Moreover, good oral availability enables the usage of nirmatrelvir/ritonavir, not only in hospitalized patients, but also among outpatients. Nirmatrelvir (PF-07321332) has been demonstrated to stop the spread of COVID-19 in animal models. Despite frequent mutations in the viral genomes of SARS-CoV-2, nirmatrelvir shows an effective antiviral effect against recent coronavirus mutants. Despite the promising antiviral effect of nirmatrelvir, there are several unresolved concerns. First, the final results of large-scale clinical trials for early therapy of mild cases of COVID-19 are not yet published. Second, the effectiveness of nirmatrelvir against upcoming variants in the coming years requires close monitoring. Considering the promising preliminary results of the EPIC-HR trial, nirmatrelvir/ritonavir in conjunction with vaccines and non-pharmacological interventions, may represent the dawn in the dark of the COVID-19 pandemic.

## 1. Introduction

Coronavirus disease 2019 (COVID-19), due to severe acute respiratory syndrome coronavirus 2 (SARS-CoV-2), has resulted in a global outbreak spread by respiratory transmission [1,2,3]. Clinical presentations of COVID-19 range from no, or mild, symptoms, to severe hypoxia, respiratory or multiorgan failure, and even death [2]. The SARS-CoV-2 pandemic has resulted in 5.5 million deaths out of 272 million confirmed cases reported worldwide by the end of December 2021 [4]. After the alpha, beta, gamma, and delta SARS-CoV-2 variants of concern (VOCs), the most recent B.1.1.529 variant (named Omicron), classified as a VOC by the WHO on 26 November 2021, became a new threat worldwide [4]. The first case of Omicron variant was reported in Botswana, South Africa, in early November, and this variant spread quickly to 108 countries, with more than 150,000 reported cases and 26 deaths by the end of 2021 [4]. SARS-CoV-2 is expected to remain a persistent global threat in 2022.

SARS-CoV-2 is composed of four structural proteins, including spike glycoproteins (S), small envelope glycoproteins (E), glycoprotein membranes (M), nucleocapsids (N), and other accessory proteins [5]. The spike protein of SARS-CoV-2, a transmembrane protein with angiotensin-converting enzyme 2 (ACE2) as the receptor for cell entry, has been regarded as a surrogate target for vaccine development [5,6].

Vaccines are considered to be the most significant and effective advance against the COVID-19 pandemic [7]. In addition to vaccines, many new drugs have been investigated or approved for combating COVID-19, such as remdesivir, molnupiravir, and nirmatrelvir/ritonavir [7,8,9,10,11]. Remdesivir (RDV; GS-5734) is currently the only FDA-approved antiviral drug for the treatment of SARS-CoV-2 infection. However, RDV must be administered intravenously, restricting its clinical use to patients with relatively advanced disease requiring hospitalization [8]. Molnupiravir, an inhibitor of RNA-dependent RNA polymerase, provided a 30% reduction in hospitalization or death compared with the placebo group in a phase III study enrolling non-hospitalized adults with mild to moderate COVID-19, and at least one risk factor for severe disease, within five days of symptom onset [10,11]. However, molnupiravir is associated with the alarming possibilities of inducing mutations in human DNA and accelerating the development of new viral variants [10,11]. Among these new drugs, the oral form nirmatrelvir/ritonavir provides the most promising therapeutic effect, an 89% reduction in the risk of hospitalization or death [7,9,12] (Figure 1). Nirmatrelvir/ritonavir have been expected to change the course of the COVID-19 pandemic [13,14].

## 2. The Development of Paxlovid™ (Nirmatrelvir/Ritonavir)

The genome of SARS-CoV-2 includes two polyproteins, pp1a and pp1ab, and four structural proteins [2]. Polyproteins are cleaved by an important main protease (M^pro^, also referred to as 3CL protease) at 11 different sites to produce shorter, nonstructural proteins for viral replication [15]. Coronavirus M^pro^ is a three-domain cysteine protease that is characterized by a Cys145-His41 catalytic dyad at the cleft between domains I and II [12]. SARS-CoV-2 M^pro^ is critical in viral replication. From an evolutionary perspective, the amino acid sequence and 3D structure of M^pro^ are highly conserved among the *Coronavirinae* subfamily [15]. Examination of M^pro^ sequences between known SARS-CoV-2 variants have showed a low mutation frequency [15]. In the era of emerging variants, the inhibition of SARS-CoV-2 M^pro^ has become a potential target for antiviral therapy to treat COVID-19 [12].

Nirmatrelvir (or PF-07321332)/ritonavir, developed by Pfizer, Inc., is an orally bioavailable SARS-CoV-2 main protease inhibitor with extensive coronavirus antiviral activity, good off-target selectivity, and thus fewer adverse drug reactions [12]. Initially, an inhibitor of SARS-CoV M^pro^, PF-00835231, was investigated for the treatment of SARS [16]. However, PF-00835231 has the disadvantage of low passive absorptive permeability and poor oral absorption [12]. To investigate the potential of an orally bioavailable anti-SARS-CoV-2 drug, a new compound called PF-07321332 was created, based on a modification of PF-00835231, with a reversible covalent thioimidate adduct [12]. PF-07321332, now known as nirmatrelvir, showed improved oral bioavailability (F = 50%), with 95% of the oral dose absorbed from the gastrointestinal tract in rats [12]. This high oral bioavailability makes nirmatrelvir convenient for oral administration.

Nirmatrelvir has been observed to form tight conjugates with SARS-CoV-2 M^pro^ [17,18,19,20]. Analysis of the structure of SARS-CoV-2 M^pro^ in complex with nirmatrelvir shows that, in addition to the S-C covalent bond, nirmatrelvir is further stabilized through a network of hydrogen bonds and hydrophobic interactions, which further strengthens its binding to the active site of SARS-CoV-2 M^pro^ [17]. Molecular dynamics simulations provide further support for the inhibitory mechanism of nirmatrelvir toward SARS-CoV-2 M^pro^, which occurs in two steps: an initial non-covalent addition with the dyad in a neutral form with the formation of the thiolate-imidazolium ion pair, and ligand relocation for finalizing the electrophilic attack [18]. Despite the presence of lopinavir or ritonavir, nirmatrelvir still exhibited a high binding ability to the active site of SARS-CoV-2 M^pro^ [19]. The tight adherence of nirmatrelvir to SARS-CoV-2 M^pro^ makes it an efficient antiviral agent.

## 3. Effectiveness of Paxlovid™ (Nirmatrelvir/Ritonavir) In Vitro and In Vivo

Nirmatrelvir has excellent anti-SARS-CoV-2 activity in vitro [12]. Treatment of differentiated normal human bronchial epithelial cells with varying concentrations of nirmatrelvir for three days led to inhibition of SARS-CoV-2 viral replication without obvious cytotoxicity [12]. The insignificant in vitro cytotoxicity of nirmatrelvir indicates that it is a safe drug.

Based on in vitro studies, *C*YP3A4 played a primary role in the metabolism of nirmatrelvir, which suggested the possibility of boosting serum concentrations of nirmatrelvir by co-treatment with the potent CYP3A4 inactivator, ritonavir [12]. Ritonavir has been used as a pharmacokinetic enhancer of several marketed protease inhibitors of HIV (e.g., darunavir and lopinavir) that are metabolized through CYP3A4 [21]. Ritonavir is thus combined with nirmatrelvir to enhance its therapeutic concentration.

A now-commonly used mouse model of SARS-CoV-2 infection was proposed by Leist et al. [22]. In brief, 10-week-old BALB/c mice were infected intranasally with SARS-CoV-2 MA10, which led to 10% body weight loss [22]. A dose-dependent increase in morbidity and mortality over the course of 14 days with SARS-CoV-2 MA10 was noted in this model, which exhibits the disease spectrum and host immune responses of COVID-19, such as elevated T-helper (Th)-1 cytokines, and the loss of surfactant expression and pulmonary function in association with acute lung injury (ALI) and acute respiratory distress syndrome (ARDS) [22].

After infection with SARS-CoV-2 MA10, mice treated twice daily with nirmatrelvir at both 300 and 1000 mg/kg doses were protected from weight loss [12]. In addition, nirmatrelvir effectively decreased pulmonary viral loads of SARS-CoV-2 MA10 in mice [12]. Decreased inflammation of perivascular tissues and bronchial and bronchiolar epithelia was also noted among nirmatrelvir-treated mice [12]. The encouraging antiviral effect of nirmatrelvir on these mice makes nirmatrelvir a promising candidate for further clinical trials to assess its potential for the treatment of SARS-CoV-2 infection.

## 4. Therapeutic Effect of Paxlovid™ (Nirmatrelvir/Ritonavir) in Patients

In a randomized, double-blind, placebo-controlled, single-ascending-dose study in healthy adults, nirmatrelvir was safe and well tolerated, and received a significant boost in plasma concentrations if co-administered with ritonavir [12]. There are at least three ongoing clinical trials of nirmatrelvir/ritonavir for COVID-19 registered at ClinicalTrials.gov posted from July 2021 to January 2022: NCT04960202, NCT05047601, and NCT05011513 (Table 1). Of note, NCT04960202 and NCT05011513 were parts of the EPIC-HR (Evaluation of Protease Inhibition for COVID-19 in High-Risk Patients) trial, which included the cases of confirmed SARS-CoV-2 infection with the onset of COVID-19 signs/symptoms within five days, prior to randomization. The chief objective of NCT04960202 is to investigate the proportion of participants who experience COVID-19-related hospitalization or death from any cause after nirmatrelvir/ritonavir treatment. The main objective of NCT05011513 is to determine the time to sustained alleviation of all targeted COVID-19 signs/symptoms after nirmatrelvir/ritonavir treatment. NCT05047601 was used to analyze the proportion of participants who developed symptomatic, reverse transcription polymerase chain reaction (RT-PCR)-confirmed, SARS-CoV-2 infection among those who initially had a negative RT-PCR result after nirmatrelvir/ritonavir treatment. Ongoing clinical trials of nirmatrelvir/ritonavir aim to assess not only the treatment of patients with SARS-CoV-2 infection, but also the effectiveness of these drugs in preventing its spread.

According to the announcement by Pfizer, Inc., nirmatrelvir/ritonavir (Paxlovid™) significantly reduced the risk of hospitalization and death based on an interim analysis of the Phase 2/3 EPIC-HR randomized, double-blind study of non-hospitalized adults with COVID-19 and at least one underlying disease, such as diabetes or lung disease, which are risk factors for severe COVID-19 [23]. However, administering antiviral therapy to people within three days of COVID-19 is a clinical challenge, and the trial cohort was a part of a larger group who were treated within five days of symptoms [13,23]. Reductions in COVID-19-related hospitalization or death were observed in those treated within five days of symptom onset. Of patients treated with oral nirmatrelvir/ritonavir, only 0.3% (8/1039) were hospitalized during the 28 days following randomization, and none of the cases were fatal; in comparison, 6.3% (66/1046) of those treated with a placebo were hospitalized, and 12 subsequent deaths occurred; the estimated risk reduction was −6.3% (95% CI: −9.0%~−3.6%; *p* < 0.0001) [23,24]. The mortality rates for those with and without nirmatrelvir/ritonavir treatment were 0% (0/1039) and 1.1% (12/1046), respectively [24].

Therefore, the Food and Drug Administration (FDA) issued an Emergency Use Authorization (EUA) for nirmatrelvir/ritonavir for the treatment of patients: (1) with mild to moderate COVID-19 within five days of symptom onset, and (2) at a high risk of progression to severe disease, on 22 December 2021 [24]. The suggested dose for patients with normal renal function is nirmatrelvir 300 mg (two 150 mg tablets) plus ritonavir 100 mg (one 100 mg tablet) orally, twice daily, for five days [24].

The suggested dose reduction for moderate renal impairment (estimated glomerular filtration rate (eGFR) ≥30 to <60 mL/min) is 150 mg nirmatrelvir (one 150 mg tablet) with 100 mg ritonavir (one 100 mg tablet) twice daily for five days [24]. Nirmatrelvir/ritonavir is not recommended for patients with severe renal impairment (eGFR <30 mL/min) or with severe hepatic impairment (Child–Pugh Class C) [24].

## 5. Effectiveness of Paxlovid™ (Nirmatrelvir/Ritonavir) against SARS-CoV-2 Variants

Nirmatrelvir shows antiviral effects against all known human coronaviruses, including beta-coronaviruses (SARS-CoV-2, SARS-CoV-1, MERS-CoV, HKU1, and OC43) and alpha-coronaviruses (229E and NL63) [12]. Nirmatrelvir remains effective against frequently mutated SARS-CoV-2 [25,26,27]. Among the M^pro^ of five SARS-CoV-2 lineages (C.37 Lambda, B.1.1.318, B.1.2, B.1.351 Beta, P.2 Zeta), each harbors a strongly prevalent missense mutation (G15S, T21I, L89F, K90R, L205V) [25]. The extent of inhibition of nirmatrelvir against these protease variants was similar, with 5 nM of nirmatrelvir displaying inhibition activity of <50%, 20 nM showing inhibition of >50%, and 100 nM completely inhibiting the activity of protease variants of five SARS-CoV-2 lineages [25]. Treatment of Syrian golden hamsters with nirmatrelvir (250 mg/kg twice daily) could completely protect the hamsters from intranasal infection by the beta (B.1.351) or delta (B.1.617.2) SARS-CoV-2 variants [28]. Furthermore, treatment of delta-variant-infected animals with nirmatrelvir could prevent transmission to untreated cohoused sentinels [28].

The P132H mutation in nsp5 (M^pro^) has been noted in the new Omicron variant, but structural analysis reveals that this mutation does not affect the active site, and may not affect the antiviral ability of nirmatrelvir [26,27,29]. However, the real-world evidence of antiviral effect of nirmatrelvir against Omicron variants has not yet been reported.

## 6. Pharmacokinetic Properties and Safety Concerns of Paxlovid™ (Nirmatrelvir/Ritonavir)

Regarding the pharmacokinetic properties of nirmatrelvir and ritonavir in healthy subjects, the time of maximum drug concentration (T_max_) after a single dose of 300 mg nirmatrelvir (2 × 150 mg tablet formulation) administered together with a 100 mg ritonavir tablet in healthy subjects was 3.00 h and 3.98 h, respectively, and the mean half-life (t_1/2_) was 6.05 h and 6.15 h for nirmatrelvir and ritonavir, respectively [24]. Renal elimination is the major route of elimination for nirmatrelvir, whereas ritonavir is eliminated by hepatic metabolism [24].

Adverse events of nirmatrelvir/ritonavir that occurred at a greater frequency (≥5 subject difference), compared with the placebo group, included dysgeusia (6% vs. <1%), diarrhea (3% vs. 2%), hypertension (1% vs. <1%), and myalgia (1% vs. <1%) [24]. However, ritonavir used as an inhibitor of HIV-1 protease prevents the cleavage process of viral polyprotein precursors into mature and functional proteins, hence interrupting the production of new viral particles [24,30]. The common side effects of ritonavir are nausea, vomiting, diarrhea, changes in taste, fatigue, rash, and—associated with long-term use—hyperlipidemia and lipodystrophy [30,31]. Ritonavir was used to boost the effect of other antiviral drugs based on its inhibition of cytochrome P450-3A4 (CYP450-3A4) and, to a lesser extent, cytochrome P450-2D6 (CYP450-2D6) [30]. Thus, co-administration of ritonavir with other drugs metabolized by cytochrome P450 might enhance their bioavailability [24,30]. Drug–drug interactions and adverse drug reactions of nirmatrelvir/ritonavir can be expected in COVID-19 patients, as have been reported from their usage among HIV-infected patients.

In conclusion, based on the preliminary results of the EPIC-HR trial, early administration of oral nirmatrelvir/ritonavir therapy may effectively decrease the risk of COVID-19-related hospitalization or death for outpatients. With an ongoing clinical trial, the role of nirmatrelvir/ritonavir in preventing disease transmission may be further elucidated in the near future. Use of nirmatrelvir/ritonavir, an effective oral antiviral agent, together with global implementation of non-pharmacological interventions and vaccine programs, may represent the dawn in the dark of the COVID-19 pandemic.

## Figures and Tables

**Figure 1 antibiotics-11-00220-f001:**
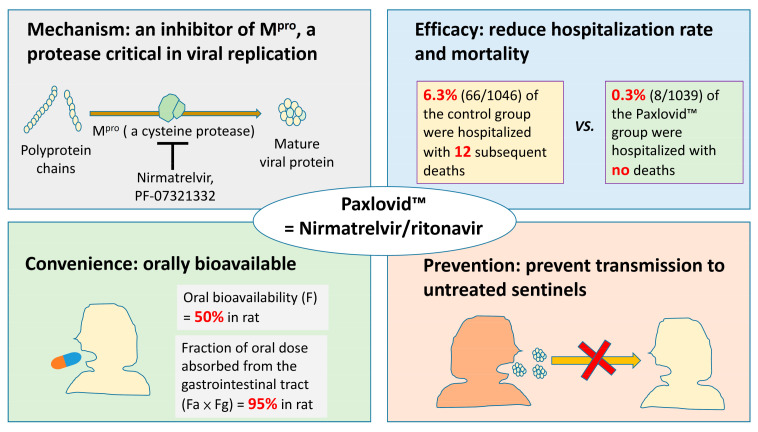
Paxlovid™ (nirmatrelvir/ritonavir), which inhibits M^pro^, may decrease hospitalization and mortality rates and prevent virus transmission, and has good oral bioavailability.

**Table 1 antibiotics-11-00220-t001:** Three clinical trials of Paxlovid™ (nirmatrelvir/ritonavir) in coronavirus disease 2019 (COVID-19) registered at ClinicalTrials.gov (accessed on 18 January 2022) posted from July 2021 to January 2022.

ClinicalTrials.gov. Identifier	Study Title	First Posted	Study Design	Included Population	Location	Outcome Measures	Status
NCT04960202	EPIC-HR: Study of Oral PF-07321332/Ritonavir Compared with Placebo in non-hospitalized High Risk Adults With COVID-19	13 July 2021	Randomized	Confirmed SARS-CoV-2 infection or initial onset of COVID-19 signs/symptoms within 5 days prior to randomization	United States	Proportion of participants with COVID-19 related hospitalization or death from any cause	Recruiting
NCT05011513	Evaluation of Protease Inhibition for COVID-19 in Standard-Risk Patients (EPIC-SR)	18 August 2021	Randomized	Confirmed SARS-CoV-2 infection or initial onset of COVID-19 signs/symptoms within 5 days prior to randomization	United States	Time to sustained alleviation of all targeted COVID-19 signs/symptoms	Recruiting
NCT05047601	A Post-Exposure Prophylaxis Study of PF-07321332/Ritonavir in Adult Household Contacts of an Individual with Symptomatic COVID-19	17 September 2021	Randomized	Negative screening SARS-CoV-2 rapid antigen test result and who are asymptomatic household contacts with exposure within 96 h	United States	Development of a symptomatic, RT-PCR confirmed SARS-CoV-2 infection	Recruiting

## Data Availability

Data is available in a publicly accessible repository.

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
