# Peer review of "Oral Nirmatrelvir/Ritonavir Therapy for COVID-19: The Dawn in the Dark?"

_antibiotics, 2022, doi:10.3390/antibiotics11020220_

Round 1
Reviewer 1 Report
I congratulate the authors for the very interesting and topical work they have carried out. I would only suggest that they could indicate the different levels of efficacy shown by Paxlovid between against the variants published by Ullrich et al., 2021.
Author Response
Reviewer 1
I congratulate the authors for the very interesting and topical work they have carried out. I would only suggest that they could indicate the different levels of efficacy shown by Paxlovid between against the variants published by Ullrich et al., 2021.
Reply: The extent of inhibition of Paxlovid across different protease mutants of five SARS-V oV-2 lineages was clarified (line 181-184).
Reviewer 2 Report
Overall the manuscript is very well written and will interest the readers, considering oral antiviral therapy review articles are needed to understand COVID19 management better.
Materials and Methods: review article, and I am satisfied with the review process.
Style of writing: English language usage and grammatical errors are non-existent.
My suggestions to the authors: if they can also include the current antivirals and then mention the role of paxlovid, it can add more weightage to the article.
Author Response
Reviewer 2
Overall the manuscript is very well written and will interest the readers, considering oral antiviral therapy review articles are needed to understand COVID19 management better.
Materials and Methods: review article, and I am satisfied with the review process.
Style of writing: English language usage and grammatical errors are non-existent.
My suggestions to the authors: if they can also include the current antivirals and then mention the role of paxlovid, it can add more weightage to the article.
Reply: Current antiviral agents were mentioned in line 57-67, and the higher treatment efficacy of Paxlovid, as compared to that of molnupiravir, was highlighted in line 68-69.